# An Assessment of Streambank Erosion Rates in Iowa

Keith E. Schilling [1,*], Calvin F. Wolter [2], Jason A. Palmer [2], William J. Beck [3], Forrest F. Williams [3], Peter L. Moore [3] and Thomas M. Isenhart [3]

[1] Iowa Geological Survey, University of Iowa, Iowa City, IA 52242, USA
[2] Iowa Department of Natural Resources, Des Moines, IA 50319, USA
[3] Department of Natural Resource Ecology and Management, Iowa State University, Ames, IA 20011, USA
* Correspondence: keith-schilling@uiowa.edu; Tel.: +1-319-335-1422

**Abstract:** Streambank erosion is a major contributor to watershed suspended sediments and phosphorus exports in many regions, but in Iowa and other midwestern states, the load contribution from streambanks is not considered by state nutrient reduction strategies. The study's objectives were to evaluate the annual bank erosion rates measured in Iowa using erosion pins and aerial imagery and assess how recession rates vary across space, time, and stream order. The overall goal was to determine whether there are global similarities to these streambank recession rates that could be generalized and scaled up for regional assessments using data from Iowa-based erosion pin studies and original research on stream migration rates. At the erosion pin sites, the recession rates averaged approximately 11 cm yr$^{-1}$ in third-order streams and, when combined with stream migration analyses, we observed scaling associated with bank recession rates at longer time scales across a range of stream orders. More bank recession occurs in larger streams and rivers with greater discharge from larger watershed areas and an increase in stream power. Variations in these bank recession rates were observed in Iowa landform regions mainly due to differences in geology and the composition of the streambank sediments. The study's results provide a temporal and spatial context for evaluating streambank recession in Iowa and the glaciated Midwest.

**Keywords:** streambank; erosion; suspended sediment; phosphorus; water quality





## 1. Introduction

Streambank erosion is a major contributor to total watershed suspended sediment exports in many regions, e.g., [1–6], with streambank contributions to annual sediment loads ranging from <10 to 96%, e.g., [3,7–12]. Streambank erosion has also been identified as a major source of riverine phosphorus (P) exports [6], with P loads from streambanks ranging from 15–93% in Denmark [13,14]; 7–10% in Minnesota [2]; 31–100% in Oklahoma [4,15]; 6–30% in Vermont [16]; to 3–38% in Iowa [17]. Overall, streambank erosion is characterized by a high degree of variability across a range of spatial and temporal scales [5,6].

Streambank erosion mechanisms include subaerial processes, freeze–thaw, shear-driven erosion, and mass wasting, and these can differ within the same watershed and across multiple watersheds [18–20]. At the scale of individual banks, the dominant erosion processes have been delineated (e.g., [21–23]), but these processes are difficult to generalize to larger scales. Wilson et al. [20] reported that the causal factors contributing to the variations in the streambank erosion in Iowa include soil texture, bulk density, seasonal soil moisture, and freeze–thaw events.

In Iowa, as in other midwestern U.S. states, efforts are underway to reduce nutrient loading through the implementation of various nutrient reduction strategies [24,25]. Notably absent from many of these strategies is an estimate of the load contribution from streambanks due to the time and effort needed to collect the bank recession data and the high degree of spatial and temporal variability in the results [17]. Various methods have been used to evaluate the rates of streambank erosion in Iowa watersheds, including

cross-section surveys, bank erosion pins, and photogrammetric methods including the use of Light Detection and Ranging (LiDAR) systems and sequential aerial photographs [6,25]. Beck et al. [17] evaluated how a 17% increase in a channel cross-section area over a 16-year period reduced the flux of sediments and P to the floodplain in a southern Iowa watershed. In low-order Iowa streams, erosion pins have been widely used to measure the bank recession at targeted eroding bank segments, e.g., [5,17,26,27]. Such erosion pin studies provide much needed detail on the timing of erosion events, the spatial variability in the recession rates among streambanks [5], and the impacts of riparian land use on streambank and gully erosion [28,29].

In contrast to bank-specific measurements, LiDAR and other remote data acquisitions provide assessments of river bank erosion at much larger scales. Streambank movement was documented in the South Fork watershed in north central Iowa using detailed mapping of the channel morphology [30,31]. Tomer and Van Horn [30] showed that the stream channels in the watershed widened from 0.5 to 1.1 m in response to the 2008 flood. Likewise, in southern Minnesota, Thoma et al. [9] and Kessler et al. [32] also used LiDAR and helicopter surveys to assess the river bank erosion along a 56 km length of the Blue Earth River. Others have assessed the streambank erosion in Iowa with terrestrial LiDAR [33], photoelectric devices (PEEP technology; [34]), and an integration of satellite images with hydrodynamic modeling [35,36].

Evaluating streambank recession rates measured at the local and regional scales provides context for estimating the streambank contributions to watershed-scale sediment and phosphorus exports and improves our understanding across a range of environmental conditions. Given the significance of streambank erosion in Iowa and the U.S. Midwest, we ask whether there are global similarities or patterns among these bank recession rates that could be generalized and scaled up for regional assessments. The objective of this paper is to report on the annual bank erosion rates measured in Iowa using both erosion pins and aerial imagery mapping and to assess how recession rates vary across space, time, and stream order. Although we focus on Iowa, the study's results could be applicable to other agricultural regions in the U.S. Midwest with similar agricultural hydrology [37].

## 2. Materials and Methods

### 2.1. Regional Setting

The state of Iowa is the largest producer of corn (*Zea mays* L.) and soybeans (*Glycine max* [L.] Merr.) in the U.S. [38], owing, in part, to its organic-rich, glacial-derived soils and favorable climate for rain-fed crop production. The surficial geology of Iowa is dominated by Pleistocene glacial deposits consisting of fine-textured glacial till and loess of varying ages [39]. In this study, we evaluated differences in bank recession rates using Major Land Resource Areas, or MLRAs. These areas are geographically associated land resource areas delineated by the Natural Resources Conservation Service for characterizing regions based on soils, landscape, precipitation, and temperature [38]. Iowa is part of 10 MLRAs and we considered the 8 major MLRAs in our analysis.

The Wisconsin-age Des Moines Lobe represents the most recent glacial advance into Iowa from around 15,000 years ago (MLRA 103—Central Iowa and Minnesota Till Prairies; Figure 1). The low-relief topography of the region stands in contrast to the hillslope-dominated terrain found in the western and southern parts of the state. The MLRA region 107B (Iowa and Missouri Deep Loess Hills) is dominated by thick loess deposits, whereas the MLRA regions 108C (Illinois and Iowa Deep Loess and Drift, West-Central Part), 108D (Illinois and Iowa Deep Loess and Drift, Western Part), and 109 (Iowa and Missouri Heavy Till Plain) in southern and southeast Iowa consist largely of rolling landscapes of thin loess overlying pre-Illinoian till. The topography of the MLRAs 104 (Eastern Iowa and Minnesota Till Prairies) and 107A (Iowa and Minnesota Loess Hills) is less sloping due to loess cover over recent glaciation (107A) and extensive erosion (104). The landscape and river corridors of the MLRA region in northeast Iowa (105—Northern Mississippi Valley Loess Hills) are dominated by thin soils overlying fractured Paleozoic bedrock. Together,

the landscape diversity of Iowa mirrors the diversity of the agro-hydrologic regions found throughout the upper Mississippi and Ohio river basins [37].

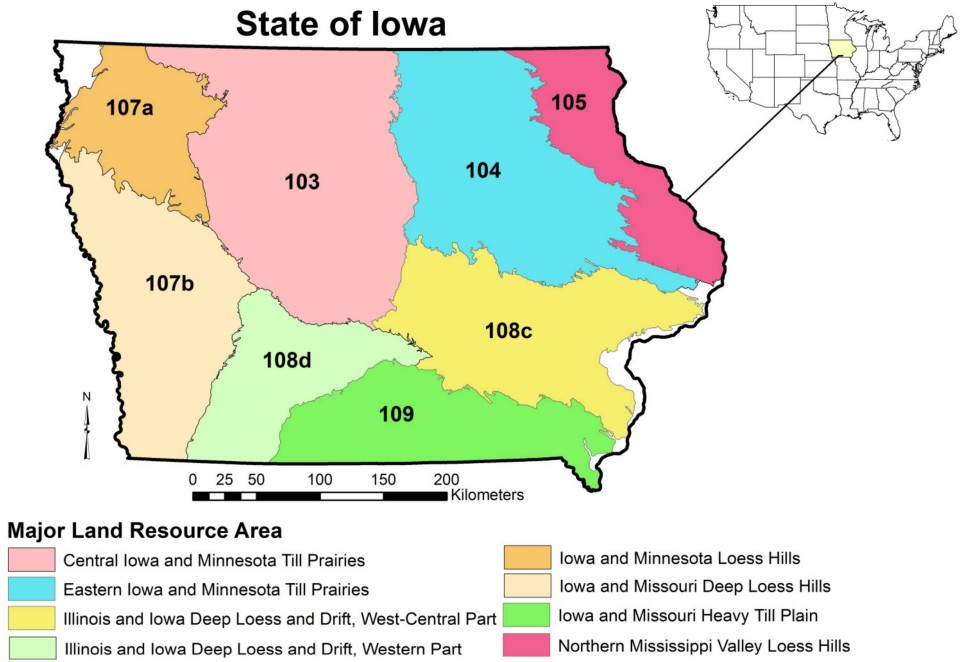

**Figure 1.** Location of MLRA regions in Iowa delineated by the USDA [38].

Iowa has a humid continental climate with hot and humid summers and cold and relatively dry winters. Its average annual precipitation and temperature range from approximately 700 to 900 mm and 6 to 12 C across a gradient from northwest to southeast Iowa [18].

*2.2. Erosion Pins*

Erosion pins have been used by several researchers to assess the magnitude of the bank recession in the 3rd- to 4th-order streams of Iowa and these methods have been reported by the studies' authors [5,17,26,27,40,41]. In general, the erosion pin methodologies employed were similar among the studies since they were conducted as projects led by researchers in the Department of Natural Resource Ecology and Management at Iowa State University. Briefly, stream surveys were first conducted to identify all the eroding streambanks within a watershed or study reach and a subset of the eroding banks was randomly selected for erosion pin installation. The erosion pins were generally installed along the bank face in a grid of two rows spaced vertically at 1/3 and 2/3 bank heights and horizontally one meter apart along the entire length of the selected eroding bank. Beck et al. [42] modified the arrangement of these pins to account for variations in the alluvial stratigraphy, installing pins in the midpoints of exposed stratigraphic units. Among all the Iowa erosion pin sites, the erosion pins were 762 mm long and 6.2 mm in diameter, and the exposed lengths of the pins were measured using a ruler. An increase in the exposed pin length from the previous measurement was assumed to be from bank recession, whereas a decrease in the length was assumed to indicate deposition. The frequency of the pin measurements varied among study sites and projects, but for this study, we compiled the Iowa pin data into annual erosion rates. In total, the annual bank recession data measured using erosion pins were available for 385 streambanks in Iowa (Table 1).



**Table 1.** Summary of annual streambank recession rates measured using erosion pins at various Iowa sites. Most studies included repeat measurements of the same sites over the years of monitoring; variations in number of banks in [36] were due to logistics in site visits.

| Study | Region of IOWA | MLRA | Years of Monitoring | No. of Banks in Study | Recession Rate (cm yr$^{-1}$) |
|---|---|---|---|---|---|
| Beck [42] | Southern | 108c | 2016 | 10 | 12.3 |
| | | | 2017 | 10 | 6.3 |
| | | | 2018 | 10 | 18.6 |
| Williams [41] | Central | 103 | 2011 | 28 | −1.23 |
| | | | 2012 | 35 | −0.40 |
| | | | 2013 | 34 | 3.90 |
| | | | 2014 | 35 | 4.63 |
| | | | 2015 | 33 | 21.36 |
| | | | 2017 | 25 | −0.50 |
| | | | 2018 | 24 | 30.97 |
| Palmer et al. [5] | Southern | 108c | 2005 | 10 | 0.4 |
| | | | 2006 | 10 | −0.6 |
| | | | 2007–2008 | 10 | 19.2 |
| | | | 2009 | 10 | 34.2 |
| | | | 2010 | 10 | 27.0 |
| | | | 2011 | 10 | 13.6 |
| Tufekcioglu et al. [26] | Southeast | 109 | 2006 | 13 | 11.7 |
| | | | 2007 | 13 | 26.6 |
| | | | 2008 | 13 | 26.3 |
| Zaimes et al. [27] | Central | 103 | 2001 | 5 | 10.3 |
| | | | 2002 | 5 | 9.5 |
| | | | 2003 | 5 | 20.2 |
| | Northeast | 104 | 2001 | 4 | 5.8 |
| | | | 2002 | 4 | 9.2 |
| | | | 2003 | 4 | 11.9 |
| | Southeast | 109 | 2001 | 5 | 8.6 |
| | | | 2002 | 5 | 2.2 |
| | | | 2003 | 5 | 15.1 |
| | | | | Average | 12.4 |
| | | | | St dev | 10.3 |
| | | | | Median | 11.0 |

*2.3. Aerial Imagery Analysis of Stream Migration*

To expand upon the reach-scale erosion pin estimates, the long-term annual average streambank recession rates in the 3rd- to 6th-order streams of Iowa were estimated using changes in the stream migration occurring over a 25-year period. Color infrared (CIR) photographs are available for Iowa from the early 1980s and approximately 2010 (note that the dates on the photographs are slightly different across the state) and the changes in the streambank locations between these two time periods were used to estimate the long-term bank migration. The 2010 CIR photos were 1:4800 in scale, whereas the 1980s photos were flown at a 1:58,000 scale. The 2010 photos were used for the horizontal control in the 1980s orthorectification. The error for the 1980s photos is unknown because it varied around the state, depending on what control points were available.

We selected a subset of stream reaches in the 3rd- to 6th-order streams of each MLRA to measure the stream migration. The stream segments were created by dividing the stream lengths into segments corresponding to 30 times the average channel width, in order to account for full meander belts. From this population, a subset of 100 stream centerline segments was randomly selected from the 3rd- to 6th-order streams of the eight MLRA regions from the statewide stream segment coverage. For each selected segment, a stream centerline was digitized from the 1980 and 2005 aerial images, starting and ending at the

segment's endpoints. The width of the channel was assumed to be constant for digitizing the stream centerline. The two-line segments were converted into polygons and the area between the lines was calculated as the area over which the stream segment migrated over the 25-year period. Examples of the analyses for the typical 3rd-order and 6th-order streams are shown in Figures 2 and 3, respectively. The average rate of migration over the 25-year period for the segment was estimated by dividing the polygon area by the segment length. The annual bank recession rate was determined by dividing the long-term value by 25. In total, the stream migration was estimated in 2111 stream segments (Table 2). Although we analyzed a subset of 100 stream segments in each MLRA, some streams were obscured by overhanging vegetation or flooding and the migration rates could not be determined. Hence, the number of values (count) reported in Table 2 are less than 100.

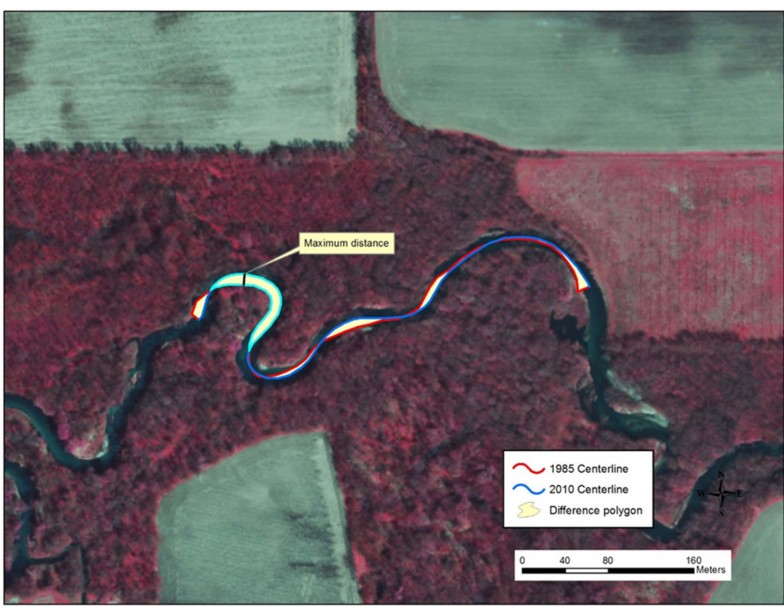

**Figure 2.** Stream migration measured in a third-order watershed (base map is 2010 photo). The maximum distance refers to the portion of the stream with the greatest change in stream centerline and maximum recession.

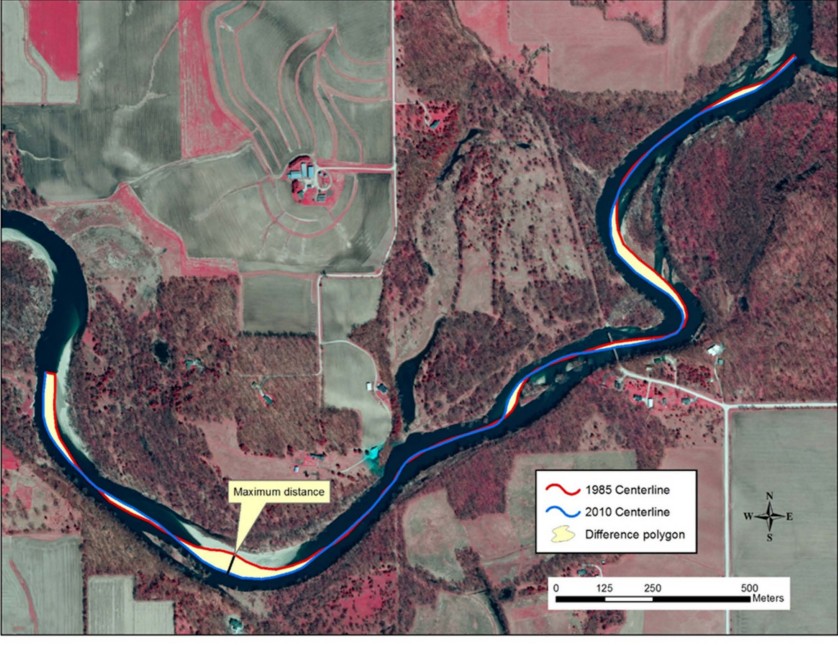

**Figure 3.** Stream migration measured in a sixth-order watershed (base map is 2010 photo).

**Table 2.** Summary of annual streambank recession by stream order and MLRA estimated using changes in channel morphology over 25 years. Count refers to the number of individual stream reaches evaluated in each MLRA and stream order. Stream segments lengths varied by stream width (e.g., segment lengths were 30 times average width).

| | Mean Rate (cm yr$^{-1}$) | | | | | | | | | | | |
| MLRA | Order 3 | | | Order 4 | | | Order 5 | | | Order 6 | | |
| | Mean | Std Dev | Count | Mean | Std Dev | Count | Mean | Std Dev | Count | Mean | Std Dev | Count |
|---|---|---|---|---|---|---|---|---|---|---|---|---|
| 103 | 12.8 | 8.3 | 74 | 16.2 | 7.7 | 81 | 36.5 | 32.6 | 64 | 42.7 | 38.5 | 30 |
| 104 | 15.9 | 11.5 | 84 | 22.9 | 12.5 | 81 | 24.5 | 16.0 | 61 | 37.2 | 21.9 | 39 |
| 105 | 16.7 | 9.1 | 83 | 21.7 | 13.1 | 87 | 37.4 | 24.2 | 64 | 100.8 | 101.1 | 47 |
| 107a | 11.2 | 12.6 | 89 | 25.2 | 21.4 | 82 | 47.9 | 38.4 | 55 | 58.1 | 31.4 | 34 |
| 107b | 11.1 | 5.8 | 82 | 17.4 | 11.9 | 88 | 36.5 | 41.8 | 66 | 74.7 | 77.9 | 44 |
| 108c | 9.0 | 4.0 | 83 | 13.0 | 6.6 | 81 | 19.9 | 12.8 | 64 | 33.2 | 20.4 | 44 |
| 108d | 11.1 | 4.6 | 73 | 13.7 | 8.3 | 72 | 31.0 | 27.7 | 59 | 50.5 | 42.9 | 42 |
| 109 | 11.3 | 6.0 | 79 | 14.8 | 11.7 | 79 | 21.3 | 13.7 | 63 | 33.1 | 28.8 | 37 |
| average | 12.4 | 7.7 | | 18.1 | 11.6 | | 31.9 | 25.9 | 62 | 53.8 | 45.4 | |

| | Maximum Rate (cm yr$^{-1}$) | | | | | | | | | | | |
| MLRA | Order 3 | | | Order 4 | | | Order 5 | | | Order 6 | | |
| | Mean | Std Dev | Count | Mean | Std Dev | Count | Mean | Std Dev | Count | Mean | Std Dev | Count |
|---|---|---|---|---|---|---|---|---|---|---|---|---|
| 103 | 53.3 | 38.8 | 74 | 62.0 | 35.4 | 81 | 156.5 | 130.0 | 64 | 178.8 | 162.0 | 30 |
| 104 | 67.9 | 52.2 | 84 | 87.7 | 43.3 | 81 | 123.5 | 101.7 | 61 | 179.2 | 141.6 | 39 |
| 105 | 49.7 | 23.6 | 83 | 77.0 | 53.1 | 87 | 148.9 | 109.1 | 64 | 497.8 | 522.0 | 47 |
| 107a | 37.3 | 36.4 | 89 | 84.0 | 64.9 | 82 | 183.3 | 137.6 | 55 | 299.0 | 195.7 | 34 |
| 107b | 37.9 | 17.8 | 82 | 58.5 | 44.2 | 88 | 126.6 | 121.1 | 66 | 268.3 | 217.8 | 44 |
| 108c | 33.5 | 16.9 | 83 | 50.3 | 33.1 | 81 | 77.7 | 67.7 | 64 | 165.3 | 131.7 | 44 |
| 108d | 43.0 | 20.1 | 73 | 49.6 | 32.8 | 72 | 113.4 | 94.3 | 59 | 184.2 | 148.5 | 42 |
| 109 | 44.5 | 25.5 | 79 | 56.9 | 45.6 | 79 | 95.2 | 73.4 | 63 | 119.1 | 90.3 | 37 |
| average | 45.9 | 28.9 | | 65.8 | 44.1 | | 128.1 | 104.4 | | 236.5 | 201.2 | |

The differences in the erosion rates across the stream orders and 3rd-order pin measurements were performed using a single-factor ANOVA and Tukey Honest Significant Differences analysis using the R statistical software.

## 3. Results and Discussion

### 3.1. Erosion Pin Recession Rates

Since the early 2000s, several hundred streambanks across Iowa have had erosion pins installed to measure the bank recession in their wadable third- to fourth-order channels. The results from five studies led by researchers at Iowa State University are reported in Table 1, but it should be noted that the universe of pinned streambanks is actually much larger than this, as the data included in Table 1 are only for those banks with regular long-term measurements. The data included herein represents the most comprehensive set of annual streambank erosion estimates available for Iowa or similar regions in the U.S. Midwest with similar agricultural hydrology.

As might be expected, the published results from the pin studies show a wide variability in the annual recession rates, ranging from −1.2 (deposition) in central Iowa to 34.2 cm yr$^{-1}$ of bank erosion in a southern Iowa watershed. Among all the sites, an average streambank recession rate for 385 bank years was approximately 12.4 ± 10.3 cm yr$^{-1}$, with a median value slightly less than (11.0 cm yr$^{-1}$). It is important to recall that erosion pins were installed in banks that were previously identified as severely eroding using USDA-NRCS visual assessment protocols during an initial stream assessment. Hence, the mean (and median) recession rate reflects the bank retreat occurring at locations where the

streambanks were known to be eroding. The mean rate is not a river-scale rate because it is not a weighted average of all the eroding and non-eroding segments. However, the sediment contributions from the non-eroding segments are minimal relative to those from the eroding sections [43].

Walnut Creek, a third-order stream in southern Iowa (MLRA 108c), has been the subject of intense streambank erosion monitoring since 2005. Palmer et al. [5] reported that 40% of the streambanks along the main channel of Walnut Creek were considered to be severely eroding. The annual erosion pin results from the watershed are reported in both the Palmer et al. [5] and Beck et al. [42] studies. Combining the results from both studies (and the non-published gap years) provides more than a decade of streambank erosion estimates within the same watershed (Figure 4). The box plot shows the range of variation measured in the same year among 10 streambank sites, but at the same time, the plot shows annual patterns indicating little bank erosion occurring in 2005, 2006, and 2012, and significant bank erosion occurring in 2009 and 2010. This streambank erosion correlates with the local precipitation patterns [5]. All of this variation is encompassed by an overall average value of 10.7 cm yr$^{-1}$ for bank erosion occurring in the same watershed over an 11-year period. It is interesting to note that the long-term average value for Walnut Creek is similar to the average and median values for the other pin-monitored sites across the state.

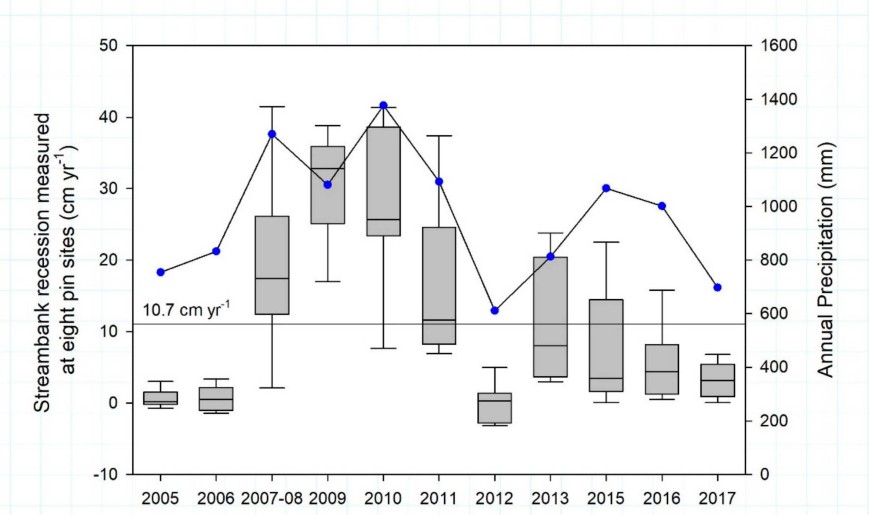

**Figure 4.** Annual streambank recession rates measured in Walnut Creek, Iowa based on data reported in [5,42]. Precipitation data downloaded from the Iowa Environmental Mesonet (https://mesonet.agron.iastate.edu/) (accessed on 1 July 2020).

Many factors account for the annual variations in the annual bank recession rates among the erosion pin sites. Iowa researchers have documented variations due to riparian land cover [44–46], cattle grazing [26], variations in precipitation and discharge [5], and alluvial stratigraphy [42], and these factors are consistent with the mechanistic processes controlling the bank erosion at individual sites around the world, e.g., [21–23,47–49]. However, some variations in the erosion pin data are also due to measurement limitations, including how to account for missing or buried pins, disturbances made during field measurements, and an underestimation of the large planar and rotational failures using pins [5,50–52]. Palmer et al. [5] remarked on the challenges of comparing these recession rates among pin sites, as dynamics reflect both local-scale and regional conditions, including variations in topography, geology, and/or climate. Despite these challenges, the average erosion rates measured using pins in Iowa converged at a long-term annual rate of approximately 11 cm yr$^{-1}$ for third-order streams. This convergence around a mean value is noteworthy because it included annual variations measured at the individual sites due



to variable climate and discharge, as well as differences in the bank recession rates among the different MLRA regions.

*3.2. Recession Rates from Aerial Imagery*

At a much larger spatial scale and over longer timeframes (25 years), the average annual bank recession rates estimated from the changes in channel positions increased with an increasing stream order in all the MLRA regions (Table 2). The mean annual recession rates increased from 12.4 cm yr$^{-1}$ in the third-order streams to 18.1, 31.9, and 53.8 cm yr$^{-1}$ in the stream orders of four through six, respectively. Single-factor ANOVAs indicated significant differences in the recession rates among the orders. All the pairwise comparisons were statistically significant, including the comparison of the third and fourth order recession rates ($p < 0.005$). The maximum recession rate represented by the maximum change in the channel migration within a channel segment was approximately 3.6 to 4.4 times greater than the mean rate (Table 2). Overall, the recession rates measured in the Iowa streams were consistent with other studies using aerial imagery, e.g., [4,47,48].

A greater mean annual recession in larger streams and rivers is consistent with greater discharge from larger watershed areas and an increase in stream power [53,54]. River scour and lateral erosion increase with stream size as a function of discharge, drainage area, and channel dimensions [55]. Hooke [53] found that watershed size explained more than 50% of the variation in the mean bank erosion rate and 39% of the variation in the maximum rates. Herein, we capture the effects of watershed size using stream order, which has a functional relation to watershed size within a hydroclimatic region [56].

To compare the erosion rates across spatial scales, it is common to normalize the erosion rate to channel width and bank height. While our analysis did not include measurements of channel width, such a normalization could be performed using our mean erosion rates by order and the existing channel geometry data from the literature. Hughes et al. [57] compiled the channel geometry data from various sources, including the national Wadable Stream Assessment (WSA) by region. Our mean bank erosion rates were normalized to that study's bankfull width estimates in the Temperate Plains ecoregion in Table 3. The resulting width-normalized mean erosion rates were similar across orders and can be interpreted to represent the erosion rates expressed as channel widths per year, e.g., [58]. This is consistent with the notion of similarity across scales in meandering river systems that has been hypothesized elsewhere [58]. This result also indicates that, for the typical third- to sixth-order streams in the state, the reach-averaged erosion rates can be estimated to fall within a narrow range of 1.2% to 1.6% of channel width per year. For example, by using this relationship, the reach-averaged erosion rate for a 10 m wide third-order stream channel would be 0.015 (/yr) × 10 (m) = 0.15 (m/yr) or 15 cm/yr. This value is similar to the third-order pin data (Table 1).

**Table 3.** Width-normalized erosion rate in Iowa 3rd- to 6th-order rivers.

| Stream Order | Bankfull Width (m) [57] | Mean Erosion Rate (cm/yr) (This Study) | Width-Normalized Erosion Rate (1/yr) |
|:---:|:---:|:---:|:---:|
| 3 | 10.4 | 12.4 | 0.0119 |
| 4 | 11.3 | 18.1 | 0.0160 |
| 5 | 24.6 | 31.9 | 0.0130 |
| 6 | 44.0 | 53.8 | 0.0122 |

In general, the recession rates were higher in the MLRA regions 104 and 105 and lower in the MLRA regions 108c and 109 (Table 2). These differences were likely due, in part, to differences in the geology and composition of the streambank sediments [59]. The loamy riparian soils of MLRA 104 and sand-dominated floodplains of northeast Iowa (MLRA 105; see [60]) are more coarse-textured than the silt- and clay-dominated streambanks of the southern and southwest Iowa areas. Channel banks with more fine particles are more cohesive, less subject to bank recession by hydraulic shear [61,62], and provide

more resistance to bank erosion [50]. The riparian soils along the lower-order streams in southern Iowa (MLRAs 108c and 109) are dominated by fine-textured sediments, although riparian zones become increasingly sand- and gravel-dominated in larger sixth-order river systems [61]. The other differences in the bank recession rates among the MLRAs may be due to factors such as width–depth ratios [53], planform geometry [62], or landscape changes such as channelization [63], agriculture [64], or urbanization [65].

Using aerial imagery allows for a long-term analysis of bank recession over much larger areas than erosion pins, although Fox et al. [6] points out that errors in the bank recession estimates could occur if the temporal or spatial resolutions of the images are not sufficient. In this study, we were not able to analyze some stream segments because the stream centerline could not be identified from the 1980s photo, due to dense vegetation or because the segment was under flood conditions. However, in some cases, the episodic changes in bank recession are best quantified using aerial imagery. For example, Tomer and Van Horn [30] analyzed the South Fork Iowa River using aerial imagery to quantify the channel changes and sediment movement from the 2008 flooding.

### 3.3. Estimating Bank Recession Rates in Iowa

In Iowa and many midwestern states, efforts are underway to reduce the nutrient loading to rivers and streams, but rarely do these strategies account for the contributions from eroding streambanks. Although streambank erosion is a natural process [61], historical row crop cultivation in the region, overgrazing, the removal of riparian vegetation, and channel widening and straightening has led to widespread channel instability and accelerated bank erosion since the turn of the 20th century [66,67]. Accurately estimating the contribution of streambanks to watershed sediment and phosphorus loading necessitates an improvement in our understanding of bank recession rates. Many states have employed various approaches to estimating the nutrient loads from point and nonpoint sources [24], but one consistent limitation has been an estimate of the nutrient load contribution from streambank erosion, which comprises a significant source of TP exports in many watersheds [42]. Developing a better understanding of bank erosion rates will help in the development of appropriate phosphorus reduction strategies for reducing these streambank contributions.

While a single appropriate bank recession rate cannot be applied to all of Iowa's stream miles, there are some generalities that can be made based on this analysis of erosion pins and stream migration rates (Figure 5). First, at the scale of individual streambanks, the bank recession rates in Iowa vary considerably on an annual scale. Based on studies using erosion pins at sites throughout Iowa, annual recession rates have ranged from net deposition to erosion in an excess of 60 cm when the pins were completely lost, e.g., [26,46]. Considering the annual rates derived from the pin studies to be mainly applicable to commonly measured third-order streams shows that a wide disparity in these annual recession rates may occur in any given year (Figure 5). This was further evidenced from the long-term annual monitoring of the erosion pins in the Walnut Creek watershed, where monitoring showed highly variable recession rates (Figure 3).

On the other hand, the aerial imagery analysis of the stream migration over a longer 25-year perspective suggests that the mean bank recession rates in Iowa scale with the stream order (Figure 5). The annual bank recession approaching 60 cm yr$^{-1}$ in larger sixth-order rivers systematically decreases to 12 cm yr$^{-1}$ in third-order streams, and an extrapolation of the trend suggests average annual recession rates of less than 10 cm yr$^{-1}$ in small first- and second-order channels. The bank recession curve intersects the pin-measured rates of the third-order channels at approximately the mean of the pin data (12.4 cm yr$^{-1}$). The convergence of the aerial imagery results with the field-measured erosion pin was supported by statistical tests (ANOVA), showing that the erosion pin data were not significantly different from the imagery-estimated recession rates at the third- and fourth-order reaches ($p > 0.1$), but were significantly different from the fifth- ($p < 0.06$) and sixth-order ($p < 0.01$) recession rates. Overall, there would appear to be some predictability of the bank recession rates at longer timeframes for Iowa's streams and rivers.

The mean pin data collected at eroding, reaching over many years, were consistent with the mean stream migration rate patterns over a 25-year period. Applying these long-term recession rates to the stream miles in Iowa highlights the importance of bank erosion in watershed-scale sediment and nutrient exports. Geomorphic changes in river systems from streambank erosion may also lead to changes in riverbed elevation and fractional sediment transport [68,69]. An important caveat to this streambank assessment is that unknown changes in the future climate could impact the stream power and bank retreat [17]. Additional changes in river hydrology from land use changes, artificial drainage, or other landscape modifications may also contribute to changes in stream power and bank recession (e.g., [10,70–74]).

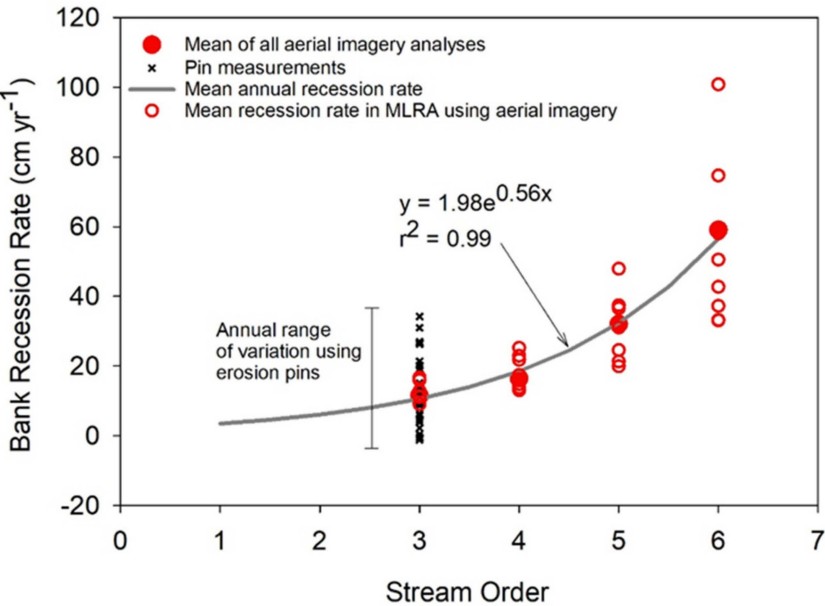

**Figure 5.** Comparison of annual pin-measured streambank recession rates measured in 3rd-order channels to the mean channel migration rates for 3rd- to 6th-order rivers estimated using aerial imagery. The regression line is reported for the mean values developed from the aerial imagery associated with each stream order.

However, it is important to note that streambank erosion is an intrinsic process in meandering alluvial streams and its effects on downstream water quality depend on whether the erosion results in a net change in sediment storage within a reach. Meander migration in a dynamically stable stream entails cutbank erosion that is approximately balanced by point-bar accretion, so that the average channel cross-section remains constant. In these cases, the net change in the sediment storage within a stream reach may be minimal. However, in dynamically unstable streams, such as those proceeding through a channel evolution sequence [58], more widespread streambank erosion can reflect this instability and lead to substantial changes in the sediment storage within a reach, exacerbating the downstream water quality and sedimentation issues. While many streams in Iowa exhibit evidence of channel evolution [62], it can be difficult to distinguish stable from unstable channel changes in short timescales. For the purpose of this analysis, the streambank erosion from all the reaches is included and it is acknowledged that the actual contribution of the streambank erosion to the suspended sediment and P exports will vary greatly among watersheds.

Although we believe that the findings from this study based on Iowa data could apply to neighboring states and the glaciated Midwest with similar agro-hydrologic landscapes [37], more work is needed to confirm the scaling relations across the region. It is clear that an improved understanding of streambank erosion is paramount to the outcomes of state nutrient reduction strategies. Many strategies fail to include streambank sources,

despite the recognition that they are potential large sources of phosphorus. As an example of this, we used the average streambank recession values reported herein to estimate the streambank contribution to the total phosphorus exports from Iowa [75]. Approximately 41% of the third- to sixth-order streambanks in Iowa were estimated to be severely eroding [76] and streambanks were found to contribute approximately 31% of the riverine TP exports from Iowa [75].

## 4. Conclusions

In this study, we compiled the streambank erosion data from Iowa erosion pin studies and conducted original research on long-term stream migration rates to assess how annual bank recession rates vary across space, time, and stream order. In a compilation of the monitoring data from the erosion pin sites, the recession rates averaged approximately 11 cm $yr^{-1}$ in the third-order streams, measured both as the long-term annual average from a single watershed and also as a compilation of the short-term recession rate measurements from the different MLRA regions across Iowa. At a much larger spatial scale and over longer timeframes (25 years), the average annual bank recession rates estimated from the changes in the channel migration increased with an increasing stream order from 12.4 cm $yr^{-1}$ in the third-order streams to 18.1, 31.9, and 53.8 cm $yr^{-1}$ in the stream orders of four through six, respectively. Overall, we found that, while annual recession rates vary considerably at the scale of individual banks, bank recession rates scale at longer time intervals across a range of stream orders. More bank recession occurs in larger streams and rivers with greater discharge from larger watershed areas and an increase in stream power. Variations in these bank recession rates were observed in the Iowa MLRA regions mainly due to differences in the geology and composition of the streambank sediments. The study's results provide a temporal and spatial context for evaluating streambank recession in Iowa and the glaciated Midwest and have implications for better understanding the challenges associated with achieving riverine phosphorus reduction goals in the midwestern U.S.

**Author Contributions:** K.E.S.: Conceptualization, methodology, writing—original draft, supervision, project administration, funding acquisition. C.F.W.: Methodology, investigation, data curation, writing—review and editing; J.A.P. methodology, investigation, data curation, writing—review and editing; W.J.B.: methodology, investigation, data curation, writing—review and editing; F.F.W.: investigation, data curation; P.L.M.: methodology, investigation, data curation, writing—original draft, writing—review and editing; T.M.I. conceptualization, writing—review, and editing. All authors have read and agreed to the published version of the manuscript.

**Funding:** Funding for the project was provided, in part, by the Iowa Nutrient Research Center under grant INRC 2017-01 and the Iowa Department of Transportation Project RB10-014 Bank Stability Assessment Tool.

**Data Availability Statement:** The data presented in this study are available on request from the corresponding author.

**Conflicts of Interest:** The authors declare no conflict of interest.

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
