# Peer review of "An Assessment of Streambank Erosion Rates in Iowa"

_environments, doi:10.3390/environments10050084_

Round 1

Reviewer 1 Report

This manuscript concerns a research paper on the field assessment of streambank erosion rates by means of erosion pins and the use of aerial imagery. The research herein presented is certainly within the scope of Environments.

However, the incompleteness and the lack of clarity in some parts of the text should be amended prior to publication. My current decision is major revision. I will be happy to review a revised version of the manuscript. Please do not take the length of comments as indicating of a negative review, as this is what I always do when I see a good article and want to help it get better.

Comments

1.       The title of the manuscript is rather vague and general. I suggest to re-write as follows: A field assessment of streambank erosion rates in Iowa.

2.       Line 61. “and the glaciated Midwest...” . This reference to glaciated areas in Iowa appears out of the blue and some contextualization is needed.

3.       Line 194. “So what accounts for the annual variations in annual bank recession rates among?”. This is a conversational style which needs re-phrasing.

4.       Figure 5. Please, include for each stream order the number of pin-measurements.

5.       Lines 313-315. These lines are ambiguous and need further explanation. Climate change is not the only factor leading to uncertainty in the future of available water resources, and ultimately, in the resulting stream power and sediment supply to rivers. For instance, decline streamflow trends are observed in natural non-managed catchments in Spain [1,2] and south of Europe [3]. This decline in water yield was mainly attributed to the generalized greenness process, as a result of land abandonment.

6.       The authors make a discussion of the results, but very biased towards previous studies carried out in Iowa. This is ok, but it would be interesting to add a couple of paragraphs comparing the results on a larger scale (i.e. including references to other regions).

7.       Lines 316-329. The implications of streambank erosion on sediment transport dynamics are more complex than described here. I think this should be mentioned and acknowledged. Large sediment particles pertaining to the riverbanks may become dislodged and thus fall into the riverbed. These large particles lead to changes in river bed elevation and in fractional sediment transport [4,5]. Riverbank erosion thus matters for suspended sediment transport and for a wide variety of river geomorphological physical processes.

8.       Lines 330-334. Remove these lines.

Bibliography

[1] Recent trends in rivers with near-natural flow regime: The case of the river headwaters in Spain. Martínez-Fernández, J. et al. PROGRESS IN PHYSICAL GEOGRAPHY. 2013.

[2] Wavelet analysis of hydro-climatic time-series and vegetation trends of the Upper Aragón catchment (Central Spanish Pyrenees). Juez et al. JOURNAL OF HYDROLOGY, 2022.

[3] Streamflow frequency changes across western Europe and interactions with North Atlantic atmospheric circulation patterns. Lorenzo-Lacruz, J. et al. GLOBAL AND PLANETARY CHANGE, 2022.

[4] How Large Immobile Sediments in Gravel Bed Rivers Impact Sediment Transport and Bed Morphology. McKie et al. JOURNAL OF HYDRAULIC ENGINEERING. 2021.

[5] Effects of bed forms and large protruding grains on near‐bed flow hydraulics in low relative submergence conditions. Monsalve, A. et al. JOURNAL OF GEOPHYSICAL RESEARCH: EARTH SURFACE, 2017.

Reviewer 2 Report

Introduction

This paper is brief, well written and well-organized. An immense amount of careful work went into collecting the datasets that the authors report and discuss. The figures are mainly good but need more extensive captions.  The principal shortcoming, which can be fixed with some work, is that the authors do not carry their regional analysis far enough for planners to know whether bank erosion is a significant source of sediment and phosphorus in Iowa watersheds.  The senior author’s research group published these kinds of data from Walnut Creek in 2018. The need for the analysis is set up in the introduction of this manuscript and by a number of the references cited. If you know the rate of bank retreat and the distribution of erosion rates in a reach, and the height of whatever is retreating (bank or terrace fill), you can calculate the tonnes of sediment released per unit reach of bank.  With some estimate of P concentration you have the loading or at least a range of possible loadings that should be refined by additional research. Are these loadings small or easily controlled compared to point source or agricultural runoff?  How much money should be spent on control?  These are the questions that planners need to weigh. The authors know this!

I note some major comments below and have added short comments, queries and suggestions to the .pdf.

1. (first paragraph of Introduction)—Here you set up the rationale for measuring erosion rates and transport of nutrients, particularly P and note that other studies have shown that bank erosion may be an important component of total erosion. In the paper you measure rates of bank erosion and stream migration and thus have the data to measure erosion rates and P transport over a wide range of streams and geologic environments, building on the work of Beck et al. (2018), but you do not? What happens to this important thread?

         2.  Erosion pins were placed  in areas of unstable banks, so presumably they sample the high end of erosion rates?  If so, how do these values compare to other banks that appear more stable? Applying a  classification system from stable to eroding is appropriate, but you need measured values or estimates to do a mass balance for a stream reach.

         3.  Figure 1 caption (see also Fig. 2 and 3 captions).  Make sure the reader has enough information! Cite the NRCS.

         4. It’d be helpful if the reader was given information on process somewhere in the Introduction.  Bank erosion in fluvial systems takes various forms and the Minnesota work suggests that moisture and freeze-thaw cycles are critical there. Is your assumption that banks erode from the surface down by dry ravel or sheetwash or as shallow mass movements rather than deeper rotational failures (slumps) that are common in glacial materials and may be hard to measure with erosion pins?

         5. Your air-photo work is impressive, but you really need to better characterize the probable error for rectification of 1980 images.  I am sure that it varies, but help the reader out since your analysis of change depends, in part, on the accuracy of your fit.

         6.  If your pins are measuring zones that have the highest rates of erosion in areas classified by NRCS criteria, how can these values (Fig. 5) overlap with mean values of stream migration measured from images over a 25-year period?

         7.  I’ve looked in vain for a characterization of how common the "active" banks are and how the locus of activity changes over time.  I am sure one answer is "it depends", but I bet the authors can do better, at least with the most carefully studied reaches.  Are the high erosion areas 5% of the stream banks....30%? If the active banks are common and activity moves along the channel with time, the pin-measured erosion rates may approach those estimated from the long-term changes in stream position.

Round 2

Reviewer 1 Report

The authorss addressed adequately my previous queries and I recommend the publication of this manuscript in present form.